# Incobotulinum Toxin Type A for Treatment of Ultraviolet-B-Induced Hyperpigmentation: A Prospective, Randomized, Controlled Trial

**DOI:** 10.3390/toxins14060417

**Published:** 2022-06-17

**Authors:** Vasanop Vachiramon, Tanaporn Anuntrangsee, Pasita Palakornkitti, Natthachat Jurairattanaporn, Sarawin Harnchoowong

**Affiliations:** Division of Dermatology, Faculty of Medicine Ramathibodi Hospital, Mahidol University, 270 Rama VI Road, Rajthevi, Bangkok 10400, Thailand; tanaporn.anunt@gmail.com (T.A.); pasita.pal@gmail.com (P.P.); natthachat.j.md@gmail.com (N.J.); hsarawin@gmail.com (S.H.)

**Keywords:** melanin, melanogenesis, neuromodulator, neurotoxin, post-inflammatory hyperpigmentation, tanning

## Abstract

Incobotulinum toxin A (IncoBoNT-A) is effective in preventing ultraviolet B (UVB)-induced hyperpigmentation. This prospective, randomized, controlled study aimed to evaluate the effect of IncoBoNT-A on the treatment of UVB-induced hyperpigmentation in 15 volunteers. Five hyperpigmentation squares (2 × 2 cm) were induced by local UVB on the abdomen at baseline. At Day 7, each site was randomized to receive no treatment (control), normal saline, or intradermal IncoBoNT-A injection with 1:2.5, 1:5, and 1:7.5 dilutions (12, 6, and 4 units, respectively). The mean lightness index (L*), hyperpigmentation improvement score evaluated by blinded dermatologists, and participant satisfaction scores were obtained at Days 21, 28, and 35. At Day 21, improvements in mean L* of 1:2.5, 1:5, and 1:7.5 IncoBoNT-A-treated, saline-treated, and control sites were 14.30%, 12.28%, 6.62%, 0.32%, and 4.98%, respectively (*p* = 0.86). At Day 28, the improvement in mean L* in IncoBoNT-A-treated groups was superior to that in the other groups. In terms of the hyperpigmentation improvement score, 12 participants (80%) experienced better outcomes with the IncoBoNT-A-injected site compared with the other sites. IncoBoNT-A, especially at higher concentrations, showed some positive effects on the treatment of UVB-induced hyperpigmentation. This may serve as an adjuvant treatment for hyperpigmentary conditions that are aggravated by UVB.

## 1. Introduction

The pigmentation of human skin is determined by several chromophores, especially melanin [1]. Melanin synthesis occurs in melanocytes, which can be triggered by the α-melanocyte-stimulating hormone (α-MSH) and adrenocorticotropic hormone activation of the melanocortin-1 receptor (MC1R), which is then transferred to epidermal keratinocytes [2]. The overproduction and accumulation of epidermal melanin production lead to common pigmentary disorders, such as melasma, freckle, post-inflammatory hyperpigmentation (PIH), and solar lentigo, which commonly affect darker skin types [3,4]. Several factors, both intrinsic and extrinsic, affect the upregulation process of melanogenesis. Intrinsic factors include fibroblasts in the dermis, hormonal and neural factors, and inflammation-related factors, whereas ultraviolet radiation constitutes the primary extrinsic factor that directly affects skin pigmentation [5].

Upon exposure to UV radiation, keratinocytes are stimulated to secrete α-MSH, which enhances melanogenesis through multiple pathways related to cAMP, protein kinase A (PKA), cAMP response element-binding protein (CREB), and microphthalmia-associated transcription factor (MITF) activity [6,7]. Several skin-lightening procedures have focused on strategies that inhibit these signals [8].

Botulinum toxin type A (BoNT-A) is a neurotoxin produced by the anaerobic bacterium *Clostridium botulinum*. This toxin is used to treat both cosmetic and non-cosmetic conditions, such as neuromuscular disorders. It acts by inhibiting the release of acetylcholine (Ach) from motoneurons’ synaptic terminals, rendering the innervating downstream structures refractory to motor nerve activation [9]. BoNT-A has been used extensively to treat various cosmetic conditions including facial dynamic wrinkle [10,11]. In 2019, Jung et al. demonstrated that BoNT-A has a beneficial effect on the reduction in ultraviolet B (UVB)-induced hyperpigmentation in both in in vitro and animal studies [12]. Subsequent research in humans demonstrated that intradermal BoNT-A injection has a preventive effect on UVB-induced skin hyperpigmentation [13].

Regarding the effect of BoNT-A injection for the treatment of UVB-induced hyperpigmentation in humans, intradermal onabotulinum toxin A (OnaBoNT-A) has yielded unsatisfactory results in a recent study [14]. This was probably due to the delayed onset of action of onabotulinum toxin A. Rappl T et al., demonstrated the different onsets of action in commercially available BoNT-A [15]. Therefore, the aim of this study was to determine the therapeutic effect of incobotulinum toxin A (IncoBoNT-A) for the treatment of post UVB-induced hyperpigmentation in human skin.

## 2. Result

### 2.1. Subjects

A total of 15 participants who met our criteria were included. The mean age was 36.93 ± 7.08 years old. Thirteen subjects (86.67%) were women, and two subjects (13.33%) were men. The Fitzpatrick classification for the skin type of 10 participants (66.67%) was type III and that of 5 participants (33.33%) was type IV. All subjects completed the study protocol and were included in the analysis. Clinical photographs are shown in Figure 1.

### 2.2. Objective Assessment

The mean L* among each experimental site showed no statistical differences at Day 0 (baseline) or Day 7 (7 days post-UVB irradiation). At Day 21 (14 days post IncoBoNT-A injection), the site injected with 1:2.5 concentration of IncoBoNT-A appeared to be the lightest area, with a mean L* of 35.35 ± 7.46, followed by the 1:5 IncoBoNT-A site (34.84 ± 7.66), 0.9% normal saline solution (NSS)-treated site (34.23 ± 7.86), control site (34.13 ± 7.39), and then 1:7.5 BoNT-A site (33.34 ± 6.98). Moreover, the 1:2.5 IncoBoNT-A site appeared to be the lightest area at each follow-up visit. At Day 21, the improvements in the mean L* of 1:2.5, 1:5, and 1:7.5 IncoBoNT-A-treated, saline-treated and control sites were 14.30%, 12.28%, 6.62%, 0.32%, and 4.98%, respectively. At the last visit (Day 35), the improvement in the mean L* of the 1:2.5 IncoBoNT-A-treated site was persistently superior to those of the others (23.38%, 14.34%, 19.60%, 9.82%, and 17.16% for 1:2.5, 1:5, and 1:7.5 IncoBoNT-A-treated, saline-treated and control sites, respectively). However, there were no statistically significant differences among these five experimental sites (*p* = 0.71). Additionally, at every follow-up visit, the mean L* of all experimental sites tended to increase over time. The mean L* at each visit is summarized in Figure 2 and Table 1.

### 2.3. Subjective Assessment

The hyperpigmentation improvement score was evaluated by a blinded physician at every follow-up visit. Twelve participants (80%) demonstrated a higher improvement score at the IncoBoNT-A-injected site. The score of all experimental sites displayed an upward pattern over time, which reflected the same pattern as mean L*. At Day 35 (28 days post-IncoBoNT-A injection), the greatest mean improvement score was graded at the BoNT-A 1:2.5 site (8.13 ± 1.27). The lowest score was graded at the 0.9% NSS site (7.05 ± 1.98). However, there were no statistically significant differences among these five treatment options (Figure 3).

### 2.4. Patient Satisfaction

The highest mean satisfaction score was observed at the control site at Day 35 (6.01 ± 2.53), followed by the 1:2.5 IncoBoNT-A site (5.88 ± 2.68), 1:5 IncoBoNT-A site (5.78 ± 2.44), 1:7.5 IncoBoNT-A site (5.42 ± 2.75), and 0.9% NSS-treated site (5.29 ± 2.73) (Figure 4).

### 2.5. Histopathologic Assessment

Histopathologic section of Fontana–Masson staining at Day 7 showed a prominent melanin deposition at the basal layer. The degree of melanin deposition was lower at D21 than at D7 at all intervention sites. However, the least amount of melanin accumulation was observed at the IncoBoNT-A 1:2.5 injection site, whereas the IncoBoNT-A 1:5 injection site had the second less amount of melanin deposition. The amounts of melanin in the control and IncoBoNT-A 1:7.5 injected sites were comparable (Figure 5).

### 2.6. Side Effects

Only burning sensation occurred in two participants (13.33%) following UVB irradiation at Day 2, which spontaneously resolved within a few days. There were no serious side effects associated with IncoBoNT-A injection. Pain associated with the injection was reported in all participants with a mean pain score of 3.7 (VAS; 0 = no pain at all and 10 = worst imaginable pain). No bruising or swelling was reported at the injection site.

## 3. Discussion

UV irradiation is a well-known cause of skin hyperpigmentation. Exposure to UV radiation (UVA and UVB) leads to an increase in reactive oxygen species (ROS) formation, resulting in skin damage. For example, exposure to UV radiation induces dermal collagen degradation, and destruction of DNA, fibrous tissue, and blood vessels of skin, leading to photoaging disorder [16,17]. As a result, various physiological responses such as epidermal hyperkeratosis, DNA repair mechanisms, antioxidant enzymes, and skin pigmentation occur to protect skin from further UV damage [18,19,20,21,22,23]. Hyperpigmentation induced by UV radiation is characterized by initial redistribution and/or molecular alterations to existing epidermal melanin pigment. Then, up-regulation of melanin production and transportation to keratinocytes occur several hours to days after UV exposure. The pro-opiomelanocortin (POMC) gene, which encodes the generation and release of α-MSH, is up-regulated by DNA and cellular damage in keratinocytes. Then, α-MSH binds to the MC1R on melanocytes in the basal epidermis, activating PKA, CREB, and MITF, all of which directly increase melanin production by stimulating the production of tyrosinase and other melanin biosynthetic enzymes [19].

At present, several treatment options have been sought by focusing on different mechanisms of hyperpigmentation, either by inhibiting melanin formation or by promoting its elimination. Several agents, including hydroxyacids, salicylic acid, and retinoic acids, have been used to stimulate epidermal turnover and desquamation in order to enhance melanin elimination [8]. Jung et al. revealed that BoNT-A also demonstrates potential in the prevention of skin hyperpigmentation induced by UVB. In their vitro study, melanocyte dendricity and melanin concentration were significantly lower at sites treated with BoNT-A. Furthermore, they conducted an experiment in mouse skin by intradermally injecting BoNT-A with a dosage of 30 units/kg 1 week before the first UVB irradiation and 1 week after the seventh UVB irradiation. As a result, several dihydroxyphenylalanine-positive melanocytes, tyrosinase activity, and melanin content were significantly lower at the BoNT-A-treated site [12]. In other human studies, BoNT-A proved to be an effective option for preventing hyperpigmentation when injecting BoNT-A 2 weeks prior to UVB irradiation [13].

BoNT-A is commonly known and aesthetically used for wrinkle treatment. It acts by inhibiting Ach release at neuromuscular junctions, resulting in transient muscle paralysis. In addition, the inhibition of Ach release at the autonomic nerve terminals results in the attenuation of glandular secretion property [24,25]. However, it was suggested that the mechanism of BoNT-A for the prevention of skin hyperpigmentation is its anti-inflammatory activity, which may affect various cytokines involved in pigmentary alterations, i.e., basic fibroblast growth factor (bFGF), interleukin 1α, and prostaglandin E2 [12,26]. Another proposed theory is that the inhibition of ACh release by BoNT-A suppresses α-MSH secretion, resulting in decreased melanin production [27].

In this study, we focused on the effectiveness of IncoBoNT-A for the treatment of UVB-induced hyperpigmentation. Our results demonstrated an improvement in mean L* among all intervention groups. However, the lightest area was produced at the IncoBoNT-A injection site with a concentration of 1:2.5 at all follow-up visits after treatment. Similar to the mean L*, the greatest hyperpigmentation improvement score at the last follow-up period (Day 35) was observed at the IncoBoNT-A 1:2.5 injection site. Although the statistical differences of these outcomes were not significant, this might be explained by the number of subjects being too small. In addition, the delay in onset of BoNT-A action may be another explanation. Typically, the clinical effects of BoNT-A begin 48 to 72 h after injection, with peak effects occurring in 2 weeks and often lasting 3 to 4 months [28,29]. However, melanogenesis occurs hours to days after UVB exposure [30,31]. Therefore, IncoBoNT-A injection at the seventh day post-UVB radiation may lead to an ineffective or partially effective inhibition of melanin synthesis. According to the histopathological evaluation, there was a decrease in melanin deposition at all intervention sites, indicating a melanin depletion process. Nonetheless, the IncoBoNT-A injected site had the lowest level of melanin at Day 21, implying that it may be involved in melanogenesis inhibition or melanin removal. The results of our study are similar to those of a recent study, in which OnaBoNT-A was used for the treatment of UVB-induced hyperpigmentation. In that study, the highest concentration of BoNT-A (i.e., 1:2.5 dilution) showed a superior result compared with the other concentrations [14]. According to a study by Rappl, IncoBoNT-A showed a faster onset of treatment compared with OnaBoNT-A (3.02 vs. 5.29 days) [15]. Therefore, treatment with IncoBoNT-A may produce a superior treatment effect due to its rapid onset. However, a comparative study with different types of BoNT-A and higher concentrations is needed to elucidate these findings. Although 12 out of 15 participants (80%) rated better improvement in hyperpigmentation at the injection site, the satisfaction score was highest at the control site. This could be explained by the side effects (i.e., pain) outweighing the benefits.

There were some limitations in our study. First, the small sample size could have affected the power of the statistical test and the generalizability of the results. Second, we investigated only UVB-induced hyperpigmentation; however, the results may alter when treating other skin hyperpigmentary conditions. Lastly, a quantitative measurement of cytokine associated with UVB-mediated skin pigmentation was not performed. Further research with a larger number of participants, a long-term follow-up period, and various hyperpigmentary conditions is required.

## 4. Conclusions

Intradermal IncoBoNT-A, especially at higher concentrations, showed some positive effects in the treatment of UVB-induced hyperpigmentation. In addition to known approved treatment conditions, it may be beneficial as an adjuvant treatment for hyperpigmentary conditions that are aggravated by UVB.

## 5. Methods

### 5.1. Study Design

This was a prospective, evaluator-blinded, randomized pilot study conducted with the approval from the Committee of Human Rights Related to Research Involving Human Subjects, Ramathibodi Hospital, Mahidol University (Protocol number MURA2021/676). The information regarding study procedures and all possible adverse effects was explained to all participants before the enrollment. The informed consent was signed by the participants before study participation.

### 5.2. Subjects

A total of 15 healthy subjects older than 18 years were included in our study. We excluded subjects who had underlying diseases, were pregnant, had active skin diseases, experienced allergic reactions to BoNT-A, had a history of photosensitive disorders, used topical medications on an ongoing basis, or had laser procedures at the experimental sites. All demographic data, including sex, age, and Fitzpatrick skin type (FPT), were recorded.

### 5.3. Treatment

The upper abdomen (5 cm above the umbilicus) was divided into 5 separate square areas of 2 × 2 cm and marked. We used a table of randomization, each of which was randomized to receive 0.3 mL of 1 of the 3 study drugs containing 12, 6, or 4 units of IncoBoNT-A, or preservative-free NSS, or no treatment, which served as a control. Each square area was intradermally injected with 0.3 mL of study drug or NSS, which was divided into 9 equal aliquots with a 30-gauge needle using a sterile technique. To prepare the study drugs, 100 units of IncoBoNT-A (Xeomin; Merz Aesthetics, Frankfurt, Germany) were diluted with 2.5 mL of preservative-free NSS. The final doses of IncoBoNT-A were achieved with this solution in amounts of 0.3 mL as-was (i.e., 1:2.5 dilution ratio), 0.15 mL was diluted with another 0.15 mL of NSS (i.e., 1:5 dilution ratio), and 0.1 mL was diluted with another 0.2 mL of NSS (i.e., 1:7.5 dilution ratio).

At baseline, we induced hyperpigmented spots by using a square-shaped tip local broadband UVB (DuaLightTM, TheraLight Inc., Carlsbad, CA, USA) at the marked 5 experimental sites. The energy for hyperpigmentation induction was based on the subjects’ skin type. Subjects with FPT skin Types III and IV were exposed to 270 and 300 mJ/cm^2^, respectively. At Day 7 after local UVB irradiation, each square area was intradermally injected with 0.3 mL of study drug or NSS at random throughout the experimental area. Subjects were instructed to avoid sun exposure, concomitant use of topical medication or whitening agent, and vigorous rubbing on the experimental site throughout the study. Follow-up appointments were scheduled at Days 21, 28, and 35 after UVB irradiation.

### 5.4. Objective Assessment

The lightness index (L*) was measured on each experimental site using a colorimeter (DSM II Colorimeter, Cortex Technology, Hadsund, Denmark) at the first visit and 4 follow-up sessions. The mean lightness index at each site was calculated from 3 repeated measurements at the same site. A comparison of the mean lightness index (mean L*) among the 5 experimental sites was evaluated.

### 5.5. Subjective Assessment

Standard digital photographs were taken at baseline and every week after UVB irradiation. Two blinded physicians evaluated hyperpigmentation improvement score from randomized digital photographs using a 10-point visual analog scale (VAS; 0 = no visible hyperpigmentation and 10 = maximal darkness). All participants also graded the satisfaction score with the same grading system. Adverse effects were recorded at every visit.

### 5.6. Histopathologic Assessment

We performed 3 mm skin punch biopsy from all 5 experimental sites in 3 willing participants at Day 7 (7 days post-UVB irradiation, before IncoBoNT-A injection) and Day 21 (21 days post-UVB irradiation or 14 days post-IncoBoNT-A injection). Skin tissues were fixed in 10% formalin and stained with hematoxylin and eosin and Fontana–Masson. Histologic analysis and the comparison of tissues were performed by 1 blinded dermatopathologist. The samples were observed under light microscopy. Three different microscopic fields per slide were photographed.

### 5.7. Statistical Analysis

All data analyses were performed using Stata Version 14.0 (Stata Corp, College Station, TX, USA). Categorical variables are presented as a frequency and percentage. Continuous variables are presented as mean and standard deviation (SD) or median and interquartile range (IQR) depending on the distribution of the data. Mean L*, physician’s improvement score, as well as patients’ satisfaction score were evaluated using linear mixed-effects models. A *p*-value less than 0.05 was considered statistically significant. Mean L* difference at Day 7 (after hyperpigmentation induction and prior to study treatment) was adjusted. *p*-value < 0.05 was considered statistically significant.

## Figures and Tables

**Figure 1 toxins-14-00417-f001:**
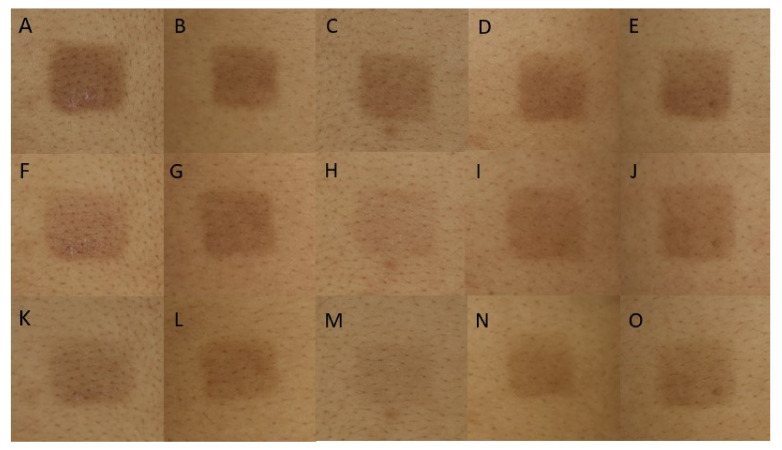
Clinical photographs of experimental sites at Day 7 ((**A**–**E**); control, 0.9% NSS, IncoBoNT-A 1:2.5, IncoBoNT-A 1:5, IncoBoNT-A 1:7.5, respectively); Day 21 ((**F**–**J**); control, 0.9% NSS, IncoBoNT-A 1:2.5, IncoBoNT-A 1:5, IncoBoNT-A 1:7.5, respectively); and Day 35 ((**K**–**O**); control, 0.9% NSS, IncoBoNT-A 1:2.5, IncoBoNT-A 1:5, and IncoBoNT-A 1:7.5, respectively). IncoBoNT-A, incobotulinum toxin type A; NSS, normal saline solution.

**Figure 2 toxins-14-00417-f002:**
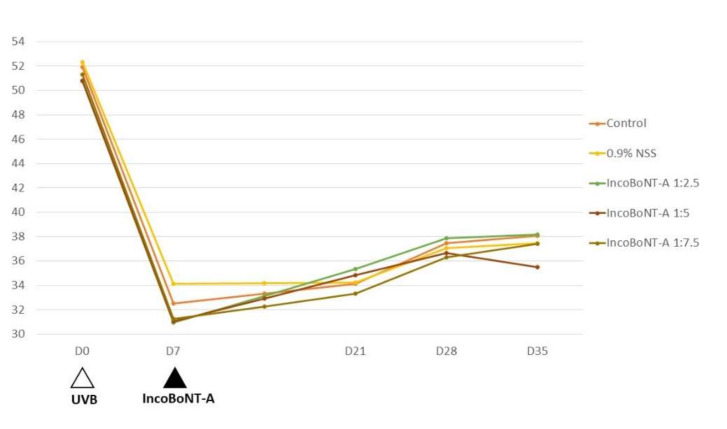
Mean lightness index by intervention group. IncoBoNT-A, incobotulinum toxin type A; UVB, ultraviolet B; NSS, normal saline solution.

**Figure 3 toxins-14-00417-f003:**
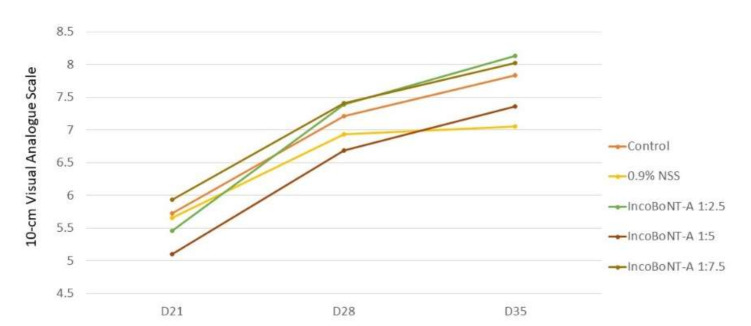
Mean hyperpigmentation improvement score rated by blinded physician by intervention group using 10 cm visual analog scale: 0 = no improvement and 10 = maximum improvement or no visible hyperpigmentation. IncoBoNT-A, incobotulinum toxin type A; NSS, normal saline solution.

**Figure 4 toxins-14-00417-f004:**
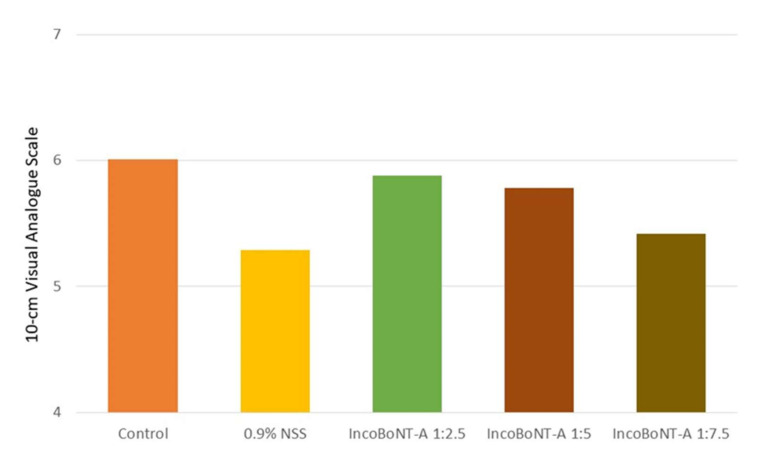
Mean satisfaction score rated by participants by intervention group using 10 cm visual analog scale: 0 = not satisfied at all and 10 = extremely satisfied. IncoBoNT-A, incobotulinum toxin type A; UVB, ultraviolet B; NSS, normal saline solution.

**Figure 5 toxins-14-00417-f005:**
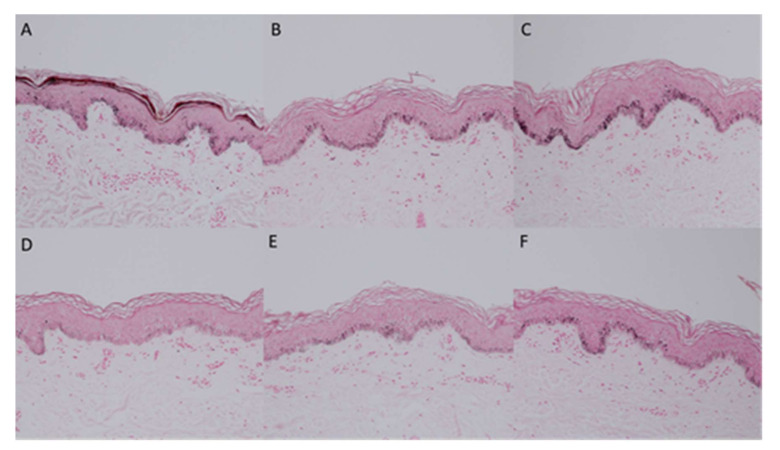
Histopathological sections of experimental sites at Days 7 (**A**) and 21 ((**B**–**F**); control, 0.9% NSS, IncoBoNT-A 1:2.5, IncoBoNT-A 1:5, and IncoBoNT-A 1:7.5, respectively) (Fontana–Masson staining, 100× original magnification). IncoBoNT-A, incobotulinum toxin type A; NSS, normal saline solution.

**Table 1 toxins-14-00417-t001:** Mean lightness index (L*).

Day	Mean L*, Mean (±SD)					
	Control	0.9% NSS	IncoBoNT-A 1:2.5	IncoBoNT-A 1:5	IncoBoNT-A 1:7.5	*p*-Value
D0 *	51.90 (±10.19)	52.34 (±6.97)	50.78 (±11.19)	50.81 (±10.39)	51.34 (±9.62)	0.900
D7 **	32.51 (±9.17)	34.12 (±9.52)	30.93 (±7.63)	31.03 (±7.31)	31.27 (±8.02)	0379
D21	34.13 (±7.39)	34.23 (±7.86)	35.35 (±7.46)	34.84 (±7.66)	33.34 (±6.98)	0.858
D28	37.49 (±8.66)	37.06 (±8.29)	37.86 (±9.32)	36.65 (±9.06)	36.28 (±7.55)	0.923
D35	38.09 (±8.25)	37.47 (±6.16)	38.16 (±6.61)	35.48 (±8.25)	37.40 (±6.68)	0.612

IncoBoNT-A, incobotulinum toxin type A; NSS, normal saline solution; SD, standard deviation. * D0: UVB irradiation. ** D7: Assigned intervention (e.g., IncoBoNT-A or NSS injection).

## Data Availability

The datasets generated during and/or analyzed during the current study are available from the corresponding author on reasonable request.

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
