# Peer review of "Incobotulinum Toxin Type A for Treatment of Ultraviolet-B-Induced Hyperpigmentation: A Prospective, Randomized, Controlled Trial"

_toxins, 2022, doi:10.3390/toxins14060417_

Round 1

Reviewer 1 Report

This is a very interesting study further supporting the role of BoNT-A in PREVENTING the post-UV hyperpigmentation in subjects with darker skin types (Asians). The manuscript is well-written, and the study design is acceptable. The results are somewhat as expected; the reviewer thinks they are clinically meaningful. Further clinical research on subjects with melasma would be very interesting.

* Some Minor Points *

[INTRODUCTION]

LINE 71. Almost 90% of participants were female.

--> Please be specific on the number of males and females.

[RESULTS]

LINES 139, 140: Only burning occurred in 2 participants (13.33%) following UVB irradiation at Day 2 after irradiation which spontaneously resolved in a short timeframe.

--> What was the degree of 'burning' after UVB irradiation? Were the two participants with 'burning' included in the study population?

[METHODS]

LINES 249, 250:  At Day 7 after local UVB irradiation, each square area was intradermally injected with 0.3 mL of study drug or NSS as random throughout the experimental area.

--> Please provide detailed information about the intradermal injection technique, i.e., the number of injection points and the injection volume per spot.

LINES 267-269: We performed 3-mm skin punch biopsy from all 5 experimental sites in 3 willing participants at Day 7 (post UVB irradiation 7 days) and Day 21 (post UVB irradiation 21 days or post IncoBoNT-A injection 14 days).

--> Was the skin biopsy on Day 7 before or after the IncoBoNT-A injection? Please specify.

LINES 270, 271: Histologic analysis and the comparison of tissues will be performed by 1 blinded dermatopathologist.

-->Please use the past tense.

Reviewer 2 Report

This is an interesting study. However, I would like to raise several concerns. 

First, this is a randomized controlled trial. Theoretically, it should be registered in an international clinical trial registry. Only an IRB approval is not enough. 

Second. only 15 patients were included. The randomization procedures were not given in details. Moreover, the sample size calculation was not provided. It is very hard to generate any conclusion from such a small number of participants. 

Third, as a randomized controlled trail, a chart elaborating the recruiting process is needed. 

Reviewer 3 Report

I have read the paper with great interest. 

Botulinum toxin (BTX-A) has a wide variety of effects and it is interesting that the authors were able to test the effect of toxin in reducing UV-induced pigmentation. It seems that a higher units of BTX-A should have been injected to the sites to properly observe the effect ot toxin.

Also, I am curious why the authors only measured the lightness (L*) of the skin and not also the erythema (a*) and melanin content (b*).

As mentioned by the authors, the observation period is also too short, since it takes months for UV induced hyperpigmentation to fade in skin type IV skin.

Round 2

Reviewer 2 Report

The article is revised according to my suggestion. 

Reviewer 3 Report

The findings do not support the conclusion that the authors suggest. Although the representative figures shows the least PIH in the botulinum toxin (highest concentration) group, there is no statistical significance in data to prove this.